# Storylines of Summer Arctic climate change constrained by Barents-Kara Sea and Arctic tropospheric warming for climate risks assessment

Xavier J. Levine[1], Ryan S. Williams[2], Gareth Marshall[2], Andrew Orr[2], Lise Seland Graff[3], Dörthe Handorf[4], Alexey Karpechko[5], Raphael Köhler[4], René Wijngaard[6], Nadine Johnston[2], Hanna Lee[7,1], Lars Nieradzik[8], Priscilla A. Mooney[1]

[1]Norwegian Research Centre, 5004 Bergen, Norway
[2]British Antarctic Survey, CB3 0ET Cambridge, United Kingdom
[3]Norwegian Meteorological Institute, 0371 Oslo, Norway
[4]Alfred Wegener Institute, 14473 Potsdam, Germany
[5]Finnish Meteorological Institute, FI-00560 Helsinki, Finland
[6]Utrecht University, 3584 CS Utrecht, The Netherlands
[7]Norwegian University of Science and Technology, 7491 Trondheim, Norway
[8]Lund University, 221 00 Lund, Sweden

*Correspondence to*: Xavier J. Levine (xale@norceresearch.no)

**Abstract**

While climate models broadly agree on the changes expected to occur over the Arctic with global warming on a pan-Arctic scale (i.e., polar amplification, sea-ice loss, increased precipitation), the magnitude and patterns of these changes at regional and local scales remain uncertain. This limits the usability of climate model projections for risk assessments and their impact on human activities or ecosystems (e.g., fires, permafrost thawing). Whereas any single or ensemble-mean projection may be of limited use to stakeholders, recent studies have shown the value of the storyline approach in providing a comprehensive and tractable set of climate projections that can be used to evaluate changes in environmental or societal risks associated with global warming.

Here, we apply the storyline approach to a large ensemble of CMIP6 models, with the aim of distilling the wide spread in model predictions into four physically plausible outcomes of Arctic summertime climate change. This is made possible by leveraging strong covariability in the climate system, associated with well-known but poorly constrained teleconnections and local processes: specifically, we find that differences in Barents-Kara Sea warming and lower tropospheric warming over polar regions among CMIP6 models explain most of the inter-model variability in pan-Arctic surface summer climate response to global warming. Based on this novel finding, we compare regional disparities in climate change across the four storylines. Our storyline analysis highlights the fact that, for a given amount of global warming, certain climate risks can be intensified while others may be lessened, relative to a "middle-of-the-road" ensemble mean projection. We find this to be particularly relevant when comparing climate change over terrestrial and marine areas of the Arctic, which can show substantial differences in their

sensitivity to global warming. We conclude by discussing potential implications of our findings for modelling climate change
impacts on ecosystems and human activities.
**1 Introduction**
Since the late twentieth century, the surface of the Arctic has warmed 2 to 4 times greater than the global average, which is
referred to as Arctic amplification (hereinafter AA, e.g., Jansen et al., 2020; England et al., 2021; Rantanen et al., 2022). This
warming amplification of the near-surface and troposphere is caused by a number of feedbacks involving oceanic, cryospheric
and atmospheric processes (Previdi et al., 2021). Sea-ice cover loss in the Arctic Ocean explains the bulk of the near-surface
warming, especially over marine areas and coastal terrestrial regions due to its impact on surface energy fluxes and upper
ocean warming (e.g., Screen and Simonds, 2010; Dai et al., 2019; Jenkins and Dai, 2021). Sea-ice loss and sea surface warming
have been singularly strong in the Barents-Kara Sea, which has been identified as a warming hotspot (Lind et al. 2018) and a
mediator of climate change between the North Atlantic and Central Arctic Oceans (Smedsrud et al., 2013). AA is also tied to
tropospheric warming, which is influenced to a greater extent by atmospheric dynamical feedback, such as temperature
feedbacks (Pithan and Mauritsen, 2014) and poleward atmospheric energy transport feedback (e.g., Merlis and Henry, 2018).
Overall, the combined influence of oceanic, cryospheric and atmospheric processes render Arctic climate change and its
surface warming amplification especially complex to predict.
AA has resulted in extensive loss of land ice, snow cover, and thawing of the permafrost over the Arctic region (e.g., Callaghan
et al., 2011; van den Broeke et al., 2016; Chadburn et al., 2017; Shepherd and IMBIE Team, 2020). These profound changes
to the Arctic climate system have been linked to increases in a range of societal and ecological risks (Yumashev et al., 2019).
For example, past decades have shown an increase in the frequency and intensity of wildfires in many Arctic regions, such as
North America's boreal forests (Masrur et al., 2018; McCarty et al., 2021), which has been attributed to unusually warm and
dry spring and summer weather conditions (Krikken et al., 2019) as well as increased lightning activity (Veraverbeke et al.,
2017). Likewise, the accelerated thawing of permafrost over large swathes of the terrestrial Arctic poses significant challenges
for the integrity of local infrastructure, such as roads and buildings (Hjort et al., 2022). Impacts of climate change in the Arctic
also extend to marine areas. For example, while increased sunlight in the photic zone from sea-ice loss and warmer sea surface
temperature may have boosted marine primary production in the Arctic oceans in past decades (Arrigo and Van Dijken, 2015),
evidence suggests that this is primarily benefiting species typically found at lower latitudes at the expense of native Arctic
species (Ingvaldsen et al., 2021). Changes to the Arctic climate system have also been suggested to have caused an increase in
the frequency and intensity of certain extreme weather over the Northern Hemisphere mid-latitudes (Cohen et al., 2014),
although the mechanisms of action and broader importance of such polar-to-midlatitude teleconnections remain controversial
(Vavrus, 2018). The loss of glaciers / land ice from Greenland, through both increased surface meltwater runoff and increased
glacier flow / dynamic ice loss, has been a major contributor to increased global sea-level rise (e.g., Rignot et al., 2011; Shepard
and IMBIE team, 2020).

Assessing the many impacts of climate change in the Arctic requires a strong understanding of the physical state of the
atmosphere, ocean, and sea ice, and how it will respond to climate change. This, however, has been hampered by future climate
projections from global coupled climate models showing a wide range of possible outcomes (Overland et al., 2019; Notz et
al., 2020; McCrystall et al., 2021; IPCC, 2021), which stems from uncertainties in possible future greenhouse gas emission
scenarios, an incomplete understanding of key climate processes and their imperfect representation in models (model
uncertainty), and natural (internal) variability within the climate system (Hawkins and Sutton, 2009). This lack of certainty
poses considerable challenges for the planning and implementation of effective mitigation strategies by stakeholders impacted
locally or remotely by Arctic climate change. The issue is often poorly addressed through the use of either a single-model or
multi-model mean climate projection (Shepherd et al., 2018).

The storyline approach overcomes the limitations of the above approaches by identifying and describing physically plausible
and self-consistent pathways that are representative of future climate change, which may be more helpful to develop mitigation
strategies (Shepherd et al., 2018). Storylines express the response of the Arctic climate to global warming conditional on a
range of environmental conditions being realised. They are based on a methodology recently developed for studying the impact
of climate change in other areas, primarily in the midlatitudes, e.g., western and central Europe (Zappa and Shepherd, 2017
[ZS17]) or Southern Hemisphere midlatitude regions (Mindlin et al., 2020 [M20]). In this study, we posit that a substantial
fraction of the variability of the surface climate response to global warming in the Arctic is associated with the warming of the
Barents-Kara Sea and the warming of the Arctic lower troposphere. This is borne out of Barents-Kara Sea warming and the
lower tropospheric warming being strongly influenced by climate variability at lower latitudes, but also being key players in
driving surface warming in the Arctic. The Barents-Kara Sea, while being sensitive to changes in the Atlantic storm track
(Jung et al., 2017) and the tropics (Warner et al., 2020), have long been recognised as a key modulators of climate variability
in Earth's Northernmost regions (Li et al., 2020; Peings et al., 2023). Likewise, the warming of the Arctic lower troposphere,
which is sensitive to changes in poleward atmospheric heat transport from lower latitudes (Russotto and Biasutti, 2020),
strongly influences the near-surface climate through its impact on the boundary layer stability and surface radiative forcing
(e.g., Previdi et al., 2020).

Using a range of possible scenarios for the Barents-Kara Sea and Arctic lower tropospheric warming that emerge from climate
model simulations, we devise storylines of future climate change for Arctic regions. Specifically, we compare the climate of
the last 30 years of the 21st century (2070–2099) projected in a high-end global warming scenario (corresponding with 8.5 W
$m^{-2}$ additional increase in radiative forcing by 2100 relative to preindustrial, the Shared Socioeconomic Pathways 5-8.5, SSP5-
8.5; see O'Neill et al. 2016 and Meinshausen et al., 2020), with the last 30 years of the historical experiment (1985–2014).
SSP5-8.5 represents the upper boundary of the range of scenarios described in ScenarioMIP and is useful to obtain the strongest
possible response to climate change within the framework of the CMIP6; this ensures that the impact of internal climate
variabilities is minimised in our study. We focus on the summer season, due to its relevance to societal and ecological impacts
at high-latitude that peak in the warm part of the year, such as, among others, high-latitude fires, trans-Arctic shipping, and
marine primary production. After describing the dataset and methodology used for our storyline analysis in section 2, we
describe in section 3 how our Arctic storylines differ from the multi-model ensemble mean response, as established by four
target variables we identified as being most relevant for studying climatic impacts in the region. We discuss the relevance of
our findings for evaluating climate impacts in the Arctic region in section 4.

## 106 2 Data and Methodology

### 107 2.1 Model data

Our analysis uses a set of 43 climate models from CMIP6, which we downloaded from The Earth System Grid Federation
(ESGF; Cinquini et al., 2014; models with members are listed on Table 1). The model and number of ensemble members
(given in parentheses) include: TaiESM1 (1), BCC-CMS2-MR (1), CAMS-CSM1-0 (2), CAS-ESM2-0 (2), FGOALS-f3-L,
FGOALS-g3 (4), (1), IITM-ESM (1), CanESM5 (15), CanESM5-CanOE (3), CMCC-CM2-SR5 (1), CMCC-ESM2 (1),
CNRM-CM6-1 (6), CNRM-ESM2-1 (5), ACCESS-CM2 (5), E3SM-1-0 (5), E3SM-1-1 (1), E3SM-1-1-ECA (1), EC-Earth3
(15), EC-Earth3-CC (1), FIO-ESM-2-0 (3), INM-CM4-8 (1), INM-CM5-0 (1), IPSL-CM6-LR (7), MIROC-ES2L (10),
MIROC6 (15), HadGEM3-GC31-LL (4), HadGEM3-GC31-MM (4), UKESM1-0-LL (5), MPI-ESM1-2-LR (15), MRI-
ESM2-0 (6), GISS-E2-1-G (14), GISS-E2-2-G (5), GISS-E2-1-H (10), CESM2 (3), CESM2-WACCM (3), NorESM2-LM (1),
NorESM2-MM (1), KACE-1-0-G (3), GFDL-CM4 (1), GFDL-ESM4 (1), NESM3 (2), CIESM (1), MCM-UA-1-0 (1). For
each model, all ensemble members of the historical experiment that were extended into the SSP5-8.5 scenario are used, capped
to a maximum of 15 members per model to limit computational resources needed to produce ensemble means for the few
models that have many members. As most models only have a few members, setting a maximum of 15 members seems a
reasonable trade-off for reducing internal variability while including as many models as possible. We find little difference in
using only a single member or an ensemble-mean of members, as the climate projections are dominated by the effect of the
climate forcing with only a small contribution from natural variability (see Fig. 1b). For each model, we produce a mean
climatology of the ensemble members for both the historical and SSP5-8.5 experiment, in their respective period of evaluation
(i.e., 1985-2014 and 2070-2099), to reduce the weight of internal variability in the climate projections. Therefore, every model
is represented by one climate projection regardless of their number of members, whether it is a single member or an ensemble-
mean of members.

## 2.2 Multivariate Linear Regression Analysis

The climate storyline approach is based on a multivariate linear regression (MLR) analysis that expresses the response to global warming of any variable, Z ("target variable"), as a linear superposition of its response to changes in $N$ climate indices, $P_i$, ("predictor index"). Following the methodology outlined in Zappa and Shepherd (2017), this can be expressed as:

$$\Delta Z(x,m) = \overline{\Delta Z}(x) + \sum_{i=1}^{N} \beta_i(x)\, \widehat{\Delta P_i}(m) \qquad (1a)$$

where $\Delta \widehat{P_i}(m) = \Delta P_i(m) - \overline{\Delta P_i}$ $\qquad (1b)$

Here, $\Delta Z$ defines changes in target variable $Z$, $\Delta P_i$ changes in predictor index $P_i$, and $\beta_i$ is the response of variable $Z$ to changes in $P_i$. Note that the target variable $Z$ varies both in space $[x]$ and across models $[m]$, but predictor indices $P_i$ only vary across models; predictor indices are typically regional averages of variables that are tied to well-known physical features of the climate. $\overline{(.)}$ defines a multi-model ensemble mean (MMM) and $\widehat{(.)}$ a deviation from the MMM; $\Delta$ defines the difference in climatology between the 2070–2099 (SSP5-8.5 emission scenario) and 1985–2014 (historical experiment) period, normalised by a global warming index, $(T_{ssp585} - T_{hist})$, i.e.,

$$\Delta X = \frac{(X_{SSP585} - X_{hist})}{(T_{SSP585} - T_{hist})} \qquad (2)$$

Here, $T$ is the annual global-mean 2 m air temperature, and $X$ defines any target variable or predictor index. Normalisation ensures that changes in target variables and predictor indices are not directly associated with changes in the global warming index ($GWI$, with $GWI = T_{SSP585} - T_{hist}$). Instead, the normalised response describes the variability in target variables or predictor indices linked to the underlying changes in the dynamics of the atmosphere/ocean/ice triggered by global warming, rather than the variability directly affected by the model's climate sensitivity.

Storylines are constructed using the coefficients $\beta_i$ emerging from the MLR analysis (Eq. 1), which are compounded with a standardised climate response for each predictor. In a 2-predictors MLR analysis, this amounts to the creation of 4 storylines that are representative of the diversity in the climate change response across CMIP6 models:

A. $\widehat{\Delta Z}_{-,+}(x) = s\,(-\beta_1(x) + \beta_2(x)\,)\,\gamma,$ $\qquad (3a)$

B. $\widehat{\Delta Z}_{+,+}(x) = s\,(+\beta_1(x) + \beta_2(x)\,)\,\Gamma,$ $\qquad (3b)$

C. $\widehat{\Delta Z}_{-,-}(x) = s\,(-\beta_1(x) - \beta_2(x)\,)\,\Gamma,$ $\qquad (3c)$

D. $\widehat{\Delta Z}_{+,-}(x) = s\,(+\beta_1(x) - \beta_2(x)\,)\,\gamma,$ $\qquad (3d)$

where $\Gamma = \frac{1}{2}\frac{1-r^2}{1-r}$ and $\gamma = \frac{1}{2}\frac{1-r^2}{1+r}.$ $\qquad (3e)$

Here, $s$ defines the standardised climate response, whose value is set to 1.26. This value is derived from a Chi-square distribution for 2 degrees of freedom and evaluated on the edge of the 80% confidence boundary region; this distribution is applied to the standardised intermodel spread in our 2 predictors from the large ensemble of CMIP6 simulations described in section 2.1. In simpler terms, $s$ defines a standardised deviation from the MMM of equal magnitude in our 2 predictor indices, which we deem plausible and yet not so extreme to be unlikely, based on the projection spread across CMIP6 simulations. To account for a weak positive correlation between both predictor indices, the storylines in Eq. (3) also contain factors $\Gamma$ and $\gamma$, which depends on the correlation coefficient $r$ (see M20 for more details).

The MLR framework of Eq. (1) and (3) seeks to predict the inter-model variability in the projections, and not the multi-model ensemble mean climate response; this is borne out of our storylines' aim, that is to explore a range of possible climate realisations representative of the diversity in model projections. While the MLR framework is compatible with using any number of predictor indices, the exponential increase in storylines with the number of predictors ($2^N$ storylines can be produced for a set of $N$ predictors) prompts us to use as few predictors as necessary, to keep the number of storylines tractable. We limit ourselves to two predictors and four storylines, as our analysis demonstrates that this configuration can explain a large fraction of the intermodel spread in the warming response of the Arctic (Table 1).

**2.3 Choice of target variables**

Due to their relevance to a broad array of climate risks, we select 2 m temperature, precipitation rate, 850 hPa zonal wind, and sea-ice fraction as target variables for understanding the impact of Arctic climate change (Lee et al., 2002). Note that the 850 hPa zonal wind is considered to be a good proxy of the near-surface wind while being less sensitive to the physical parameterization of surface processes (e.g., ZS17). This choice of variables is highly relevant to many key climate-driven risks in the Arctic, including wildfires, permafrost thawing, sea-ice loss, and marine heatwaves (Anisimov and Nelson, 1997; Pabi et al., 2008; Arrigo and Van Dijken, 2015; Melia et al., 2016). For instance, Arctic wildfires are sensitive to warm, dry, and windy conditions, which implies a dependence on near-surface air temperature, near-surface wind, and precipitation accrued during the warm season (Dowdy et al., 2010). We define 2 m temperature as our reference target variable because of its preponderance in driving those climate risks. This means that our storylines are optimised to represent the variability in the 2 m temperature.

**2.4 Choice of predictor indices**

Using the MLR approach the target variables' response to global warming may be regressed upon the two climate indices that we consider optimal for explaining differences in climate change projections between the CMIP6 model simulations. In this study, we select Arctic atmospheric amplification and Barents-Kara Sea warming as our predictors, which we refer to respectively as 'ArcAmp' and 'BKWarm'. ArcAmp is defined as the 850 hPa temperature change averaged over all areas poleward of 55° N, and BKWarm as the sea-surface temperature change averaged over the Barents-Kara Sea (its outline is

shown on Fig. 2). Both 'ArcAmp' and 'BKWarm' are defined over the extended summer season (May to October). We choose these two predictors owing to their ability to explain a large fraction of the inter-model variability in climate change projections in the Arctic, specifically the warming of the boundary layer over marine and terrestrial regions. Indeed, comparing 850 hPa temperature against surface temperature in the Arctic regions shows a strong covariability over land but weak covariability over marine areas (see Fig. 2a,b), consistent with the thermal decoupling of the marine boundary layer from the free troposphere in summer (e.g., Tjernström and Graversen, 2009). Over ocean regions, the warming of the marine boundary layer is found to warm coherently across the Central Arctic, Barents-Kara, and North Atlantic regions (Fig. 2a), in agreement with a coherent increase in sea surface temperature across those regions. Due to its role as a climate gateway between the North Atlantic and the Arctic Ocean (e.g. Smedsrud et al., 2013), we select the Barents-Kara Sea as our reference region for defining our ocean warming predictor in the Arctic. Conversely, we select the 850 hPa Arctic mean temperature warming as our second predictor due to its high degree of covariability with the warming of the terrestrial boundary layer and low degree of covariability with the marine boundary layer warming (see Table B1). The processes tying temperature anomalies in the free troposphere to those of the surface over land likely involve multiple atmospheric feedback, such as radiative or boundary layer mixing changes, which is beyond the scope of this study. Likewise, while our study leverages the connections between the North Atlantic Ocean, Barents-Kara Sea and Central Arctic Ocean warming to produce a predictor for marine boundary layer warming (see Table B2), it does not seek to identify a mechanism connecting these three regions, as it would require an in-depth analysis of changes in ocean current, upper-ocean mixing, and surface fluxes.

**3 Results**

Figure (1a) shows the intermodel spread in ArcAmp, BKWarm and GWI, which is of comparable magnitude to their MMM value for all three indices; yet we note that the spread is larger for ArcAmp and BKWarm than GWI. This large spread reflects known uncertainties in the warming of the Barents-Kara Sea and the lower Arctic troposphere in climate models, which are associated with poorly constrained physical processes and teleconnections influencing the Arctic climate (e.g., Previdi et al., 2021). Figure (1b) shows ArcAmp and BKWarm for all CMIP6 models, which shows a weak correlation in their values ($r^2 = 0.08$); this is made evident by the elliptically shaped confidence boundary region on Fig. 1b, which accounts for the larger spread in variance along the direction of correlation (the ellipticity is determined by the $\Gamma$ and $\gamma$ factors in Eq. 3). This nearly satisfies an important condition of orthogonality necessary for the effective combined use of ArcAmp and BKWarm as predictors in the MLR framework (Eq. 1). The near independence in the changes of ArcAmp and BKWarm suggests that the sensitivity of the Barents-Kara Sea and that of the lower troposphere (850 hPa) to global warming are controlled by different physical processes--even if changes in both predictor indices are ultimately driven by global warming.

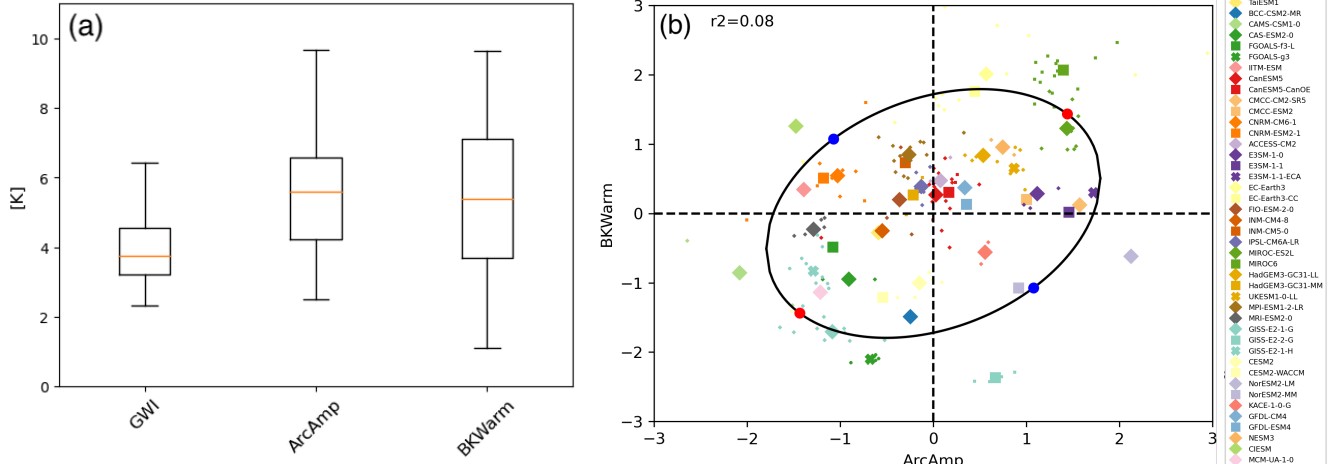

**Figure 1: (a) Boxplot showing the Global Warming Index (GWI), and the two predictor indices used for the storylines (ArcAmp and BKWarm). GWI is defined as the global and annual-mean response of the 2 m temperature, ArcAmp the response of the 850 hPa temperature averaged over all regions poleward of 55° N, and BKWarm the response of the sea surface temperature averaged over the Barents-Kara Sea (units: K). Both ArcAmp and BKWarm are defined for the extended summer season (May to October). Response is defined as the climatological-mean difference of the last 30 years of the current century (2070-2099) with that of the historical period (1985-2014). The lowest and highest values are shown at the extremities of each box; box delimiters define the 25th and 75th percentiles, while the median value (50th percentile) is shown by an orange line. (b) ArcAmp and BKWarm normalised by the GWI and with the MMM value removed for each model. Note that each predictor index is rescaled by its standard deviation, and thus non-dimensionalised (e.g., a value of 1 means a difference of one-standard deviation from the MMM value). The solid ellipse delimits the 80% confidence region of the model response in ArcAmp and BKWarm (Eq. 3). Dots on the ellipse show the 4 storylines defined in Eq. (3a-d).**

Applying the 2-predictors MLR framework described in Eq. (1), we find that the inter-model variance in the 2 m temperature explained by ArcAmp and BKWarm describes close to half of its overall inter-model variance over the Arctic (41%, see Table 1). This is about two-thirds of the theoretical maximum that can be explained using a 2-predictors MLR (64%), which we evaluated as the variance explained by the first two components of a principal component analysis (PCA) applied on the normalised change in 2 m temperature (Table 1; top row). Applying the same framework to explain changes in the 850 hPa zonal wind, precipitation rate, and sea-ice fraction, we find that the amount of variance explained by our 2-predictors MLR is substantially lower (~15%) for these variables, even if it is not insignificant. Nevertheless, evaluating the fraction of variance explained by the MLR framework on regional-scale changes (either over the Arctic or broader Northern Hemisphere high latitudes) generally indicates that our storylines have a larger explanatory power when applied to spatially coherent changes in our target variables, strengthening the relevance of our Arctic storylines to variables other than 2-m temperature (Table 1; bottom row). This highlights the fact that our storylines are tailored to quantitatively describe changes in the near-surface warming and can only provide a qualitative picture of the changes in those three variables.



|  | 2 m temperature | 850 hPa zonal wind | precipitation rate | sea-ice fraction |
|---|---|---|---|---|
| 2-PCA variance [%] | 64 | 56 | 55 | 45 |
| MLR variance [%] | 41 | 14 | 18 | 12 |
| Arctic MLR variance [%] | 68 | 35 | 33 | 11 |

**Table 1: Explained variance for 2-m temperature, sea-ice fraction and precipitation rate over the Arctic (poleward of 55° N) and**
**850-hPa zonal wind over the Northern Hemisphere high latitude regions (poleward of 40° N) in the extended boreal summer (May**
**to October), expressed as a percentage of the total variance across model projections. Each column shows a target variable. The first**
**row is the amount of variance explained by the first 2 modes of a PCA on the respective target variable, which is the maximum**
**amount of variance that could be explained by a 2-predictors MLR. The second row is the amount of variance explained by our 2-**
**predictors MLR (Eq. 1), with ArcAmp and BKWarm as predictors.  The third row is the amount of variance explained by our 2-**
**predictors MLR averaged over the Arctic (2-m temperature, precipitation rate, sea-ice fraction) and NH high latitude regions (850-**
**hPa zonal wind).**

Figure 2 shows the normalised response of each target variable in the extended summer season to each predictor index, that is
the response per degree of global warming, for a one-standard deviation in the intermodel spread of the predictor index. A
warm anomaly in the Barents-Kara Sea (BKWarm) is associated with the following: a warm anomaly in the 2 m temperature
over the Central (marine) Arctic (Fig. 2a); a dipolar anomaly in the 850 hPa zonal wind changes, with weaker winds over the
Atlantic sector of the Arctic but stronger winds over the Pacific sector (Fig. 2c); positive anomalies in precipitation rates across
all Arctic regions, especially so over land areas (Fig. 2e); and accelerated rates of sea-ice loss in the  Central Arctic, but reduced
rates of sea-ice loss the Pacific sector of the Arctic and Barents-Kara Sea (Fig. 2g). We note that sea-ice extent in the Barents
Sea region appears to be increasing in response to Barents-Kara Sea warming (Fig. 2g), a counter-intuitive finding that is likely
an artefact of the low number of models having sea-ice cover in summer in this region, as suggested by the lack of statistical
significance in the response.

These normalised response patterns strongly contrast with that associated with warm anomalies of the lower troposphere in
the Arctic (ArcAmp). For warm anomalies in ArcAmp, we find: 2 m temperature increases over most terrestrial areas (Fig.
2b); the 850 hPa zonal wind weakens over most areas around the Arctic but strengthens in the Central Arctic (Fig. 2d);
precipitation rates are reduced over most high-latitude land areas except over Greenland and the Bering Strait regions (Fig.
2f); and sea-ice loss is reduced in the Central Arctic and the Pacific sector of the Arctic basin (Fig. 2h). Both 2 m temperature
and precipitation rates response to ArcAmp are opposite to that associated with warm anomalies over the Barents-Kara Sea.
This difference in the normalised response to BKWarm and ArcAmp reflects important differences in how our two predictor
indices can modulate climate change and explain the diversity of model projections found under the SSP5-8.5 scenario
forcings.

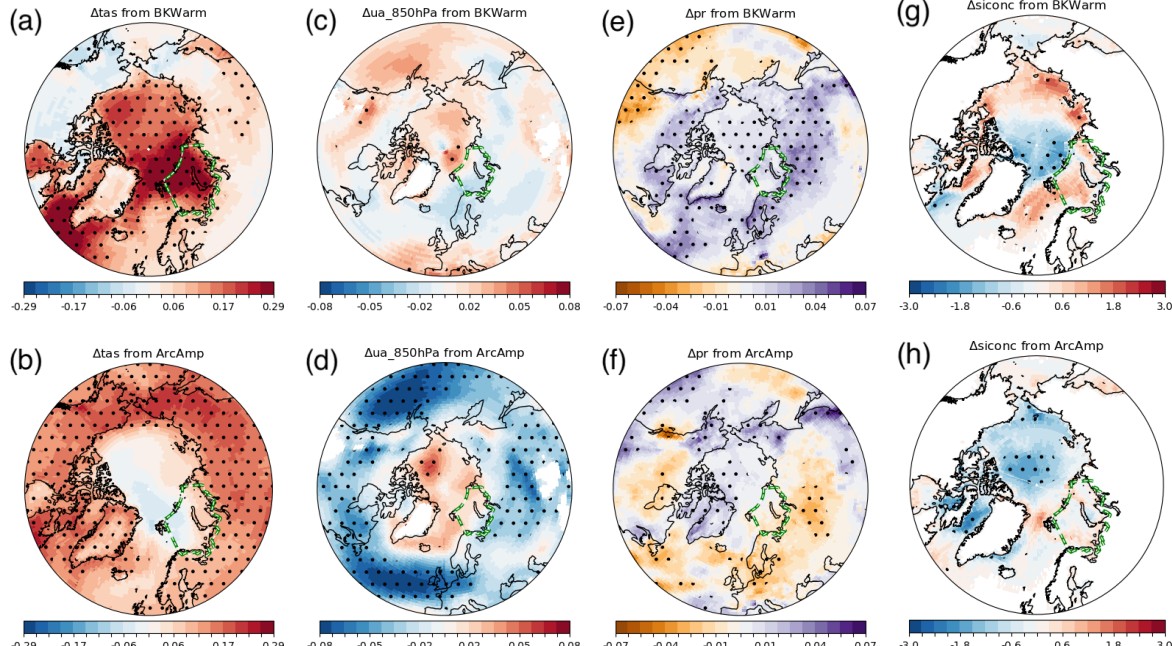

**Figure 2: Normalised response of (from left to right) 2 m temperature [K K$^{-1}$], 850 hPa zonal wind [m s$^{-1}$ K$^{-1}$], precipitation rate [mm**
**day$^{-1}$ K$^{-1}$], and sea-ice fraction [% K$^{-1}$], to a one-standard deviation in each of the predictor index for BKWarm (top row) and**
**ArcAmp (bottom row). The normalised response is the product of the regression coefficient $\beta_i$ in Eq. (1) with $\sigma_{\Delta \hat{P}_i}$, a one-standard**
**deviation anomaly in the associated predictor index. Stippling indicates statistical significance at the 95% confidence level using**
**Student's t test (i.e., p-value less than 0.05). The green dashed line delineates the outline of the Barents-Kara Sea.**

Using these normalised responses to each predictor index, we produce four storylines for each of the four target variables
according to Eq. (3). Specifically, we describe the following four storylines, referenced from A to D and defined in Eq. (3): A:
ArcAmp− / BKWarm+, B: ArcAmp+ / BKWarm+, C: ArcAmp− / BKWarm−, D: ArcAmp+ / BKWarm−. Figure 3 shows
the storylines of 2 m temperature change. First, we note that the storylines' patterns are qualitatively similar to those obtained
from the two first modes of the PCA on 2 m temperature change (compare Fig. 3a-d with A1a-d); this confirms that our
ArcAmp and BKWarm predictors capture well the dominant modes of variability that drive the intermodel spread in surface
warming projections. Consistent with the normalised response patterns (Fig. 2a-b), the main difference in 2 m temperature
between the four storylines is the rate of warming between marine and terrestrial areas of the Arctic (Fig. 3). In the MMM, the
2 m temperature is found to increase by about 1.5 to 2 K K$^{-1}$ over most oceanic and terrestrial areas of the Arctic (Fig. 3e),

showing a relative uniformity in magnitude across the Arctic. For positive anomalies in both BKWarm and ArcAmp, i.e., storyline B, the rate of warming is increased over most Arctic areas (Fig. 3b); the opposite situation is found in storyline C, i.e., negative BKWarm and ArcAmp anomalies, with a reduced rate of warming over most Arctic areas (Fig. 3c). For positive (negative) anomalies in BKWarm but negative (positive) anomalies in ArcAmp, i.e., storyline A (D), the rate of warming is increased (reduced) over marine areas but reduced (increased) over terrestrial areas when compared to the MMM (compare Fig. 3a with 3d). Changes are stronger over marine areas, especially in the northern part of the Barents-Kara Sea and the Western North Atlantic basin, where values can depart by up to 30% compared to the MMM. Out of all four storylines, storylines A and D show the largest deviation in warming rates between terrestrial and marine areas (Fig. 3a,d). Beyond an amplification or dampening of the MMM climate response, our analysis suggests a decoupling of the near-surface temperature warming between terrestrial and marine areas, with the former being associated with the lower-tropospheric warming and the latter connected to changes in the Barents-Kara and North Atlantic basin.

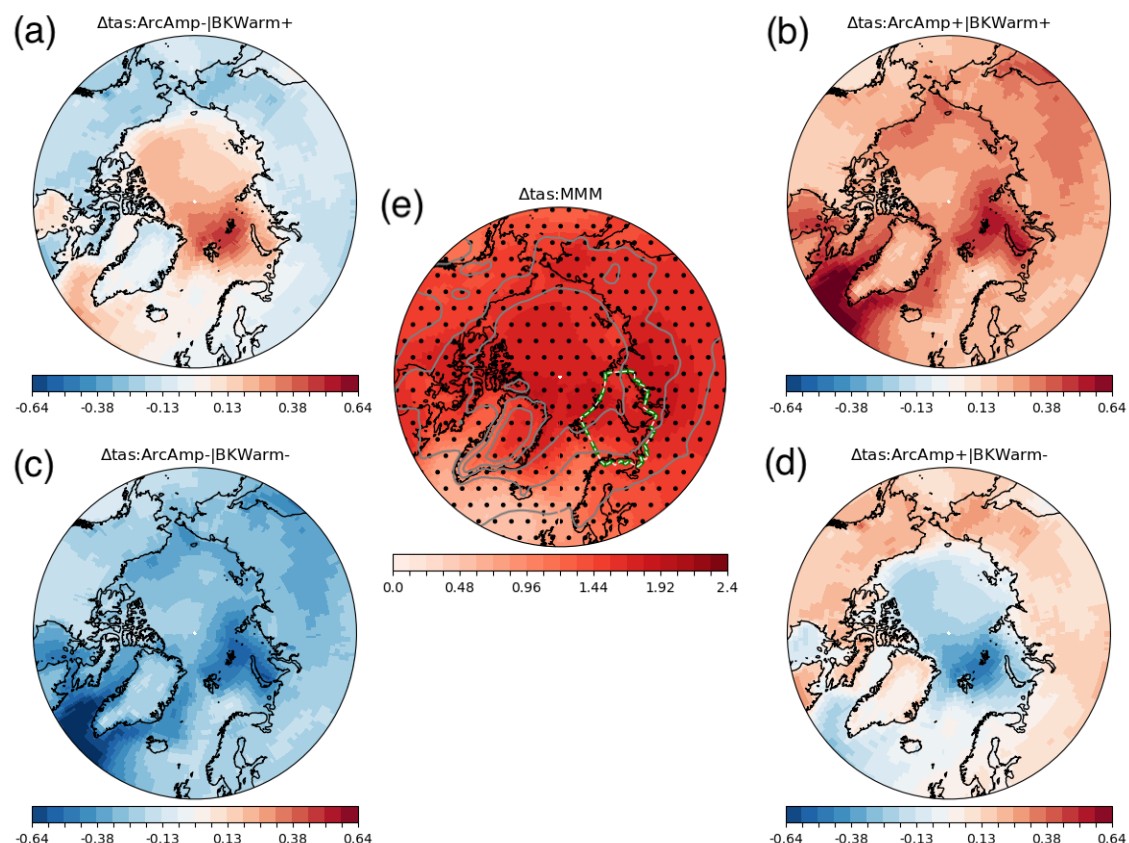

**Figure 3: (a)-(d) Storylines of climate change for 2 m temperature as defined in Eq. (3a-d) and (e) its MMM projection. Units: K K$^{-1}$. Stippling on (e) indicates areas where at least 80% of the models agree on the sign of change, and grey solid contours indicate the MMM present-day climatology. The green dashed line delineates the outline of the Barents-Kara Sea.**

In comparison with the 2 m temperature, changes in the 850 hPa zonal wind show more complexity in the spatial pattern of changes between the four storylines. In the MMM, change in the 850 hPa zonal wind (U850) shows westerly tendencies across a wide area in the circumpolar regions, spanning eastward from the Bering Sea to the Barents-Kara Sea, with a maximum over the North Atlantic between Southern Greenland and Scandinavia. The westerly tendencies extend to the Pacific sector of the Arctic Ocean, forming an arch stretching from the Beaufort Sea to the Laptev Sea. On the other hand, easterly tendencies are found in the midlatitude regions of Central Siberia. Overall, those changes suggest that in the MMM, westerly winds shift poleward and strengthen around the subpolar front and in the Central Arctic, in qualitative agreement with previously noted changes in the Northern Hemisphere mid- and high-latitude regions (Harvey et al., 2020). Going beyond the multi-model mean changes, storylines indicate a strong modulation of those changes, with storyline changes being up to 50% of the MMM. As for the 2 m temperature, storylines of U850 show modulation of the MMM response departing from a simple amplification response. Storylines B and C show a bipolar pattern (Fig. 4b,c), with easterly (westerly) tendency in the circumpolar regions but westerly (easterly) tendencies over the Arctic ocean in B (C). Likewise, storylines A and D show an apparent bipolar pattern in climate response, with changes in the subpolar regions being of opposite signs of that found in the Norwegian and Barents Sea (Fig. 4a,d). Relative to the multi-model mean changes, the poleward shift in the North Atlantic storm tracks is influenced primarily by Arctic atmospheric warming, hence linking the large uncertainty in its prediction across climate models to the intermodel spread in ArcAmp. For instance, a strengthening of the 850 hPa zonal wind in the subpolar region occurs when ArcAmp weakens, consistent with polar atmospheric warming weakening the storm tracks (e.g. Smith et al., 2019). Even if our storylines account for only a fraction of the model spread in the 850 hPa zonal wind projections, the different outcomes outlined by our storylines suggest markedly different impacts of global warming on the low-level winds, with implications for changes in synoptic storms' tracks and intensity changes.

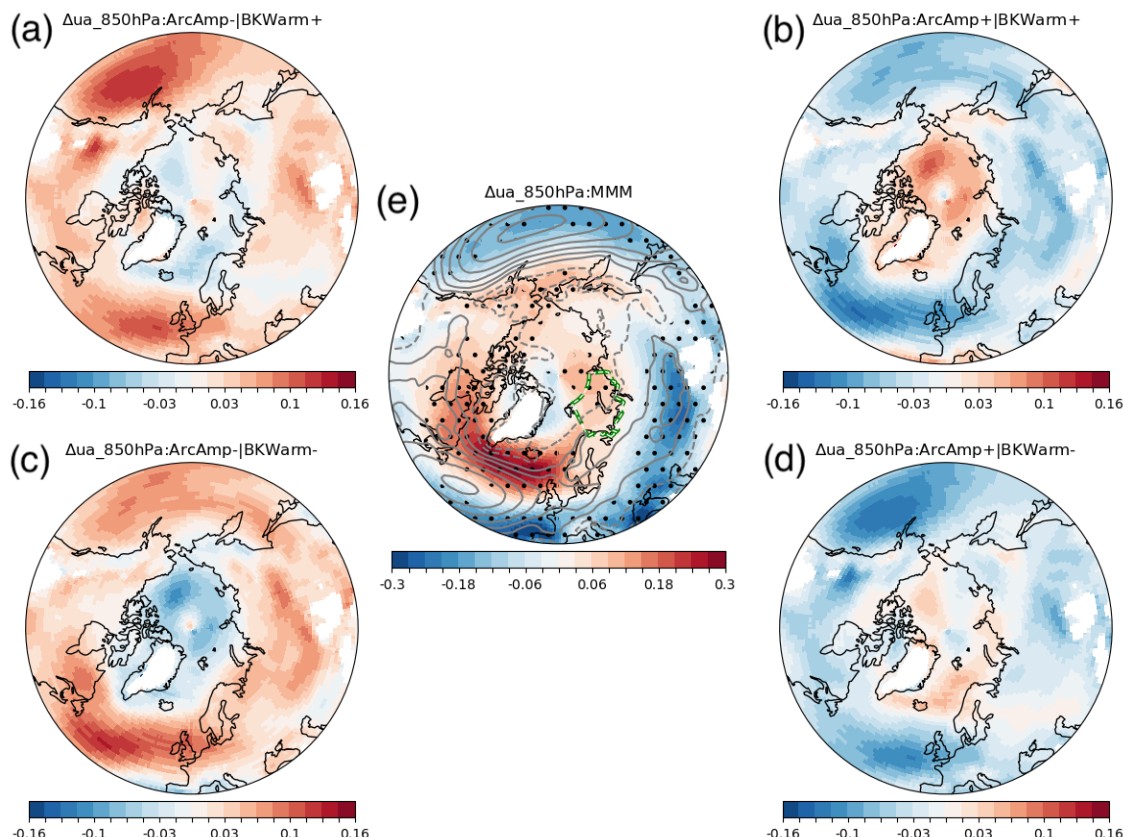

**Figure 4: Storylines of climate change for the 850 hPa zonal wind (a)-(d) and its MMM projection (e). Units: m s-1 K-1. Same convention as Fig. 3 applies.**

Figure 5 confirms the expected increase in precipitation rate changes in the high-latitude regions, in the MMM. This increase is most pronounced over mountain ranges found on the western sides of continents, which are on the paths of the Atlantic and Pacific storm tracks, e.g., the North American coastal ranges, Western Greenland, Scandinavian coastal ranges (Fig. 5e). This increase in precipitation rate contrasts with the drying tendency found over most of the midlatitude and subtropical regions of Eurasia and North America. Storylines show that projections can differ substantially from this pattern, by up to 50% of the MMM values. In particular, precipitation rate increases over most of the Arctic for positive anomalies in BKWarm (Fig. 5a,b), but decreases for negative anomalies in BKWarm (Fig. 5c,d). Changes over terrestrial areas are generally of greater amplitude than over marine areas across all storylines, and most particularly over regions of strong rainfall in the present-day climate. Overall, storylines of precipitation rates are modulated primarily by change in BKWarm, with only specific regions--notably Greenland and Siberia--showing a response to ArcAmp.

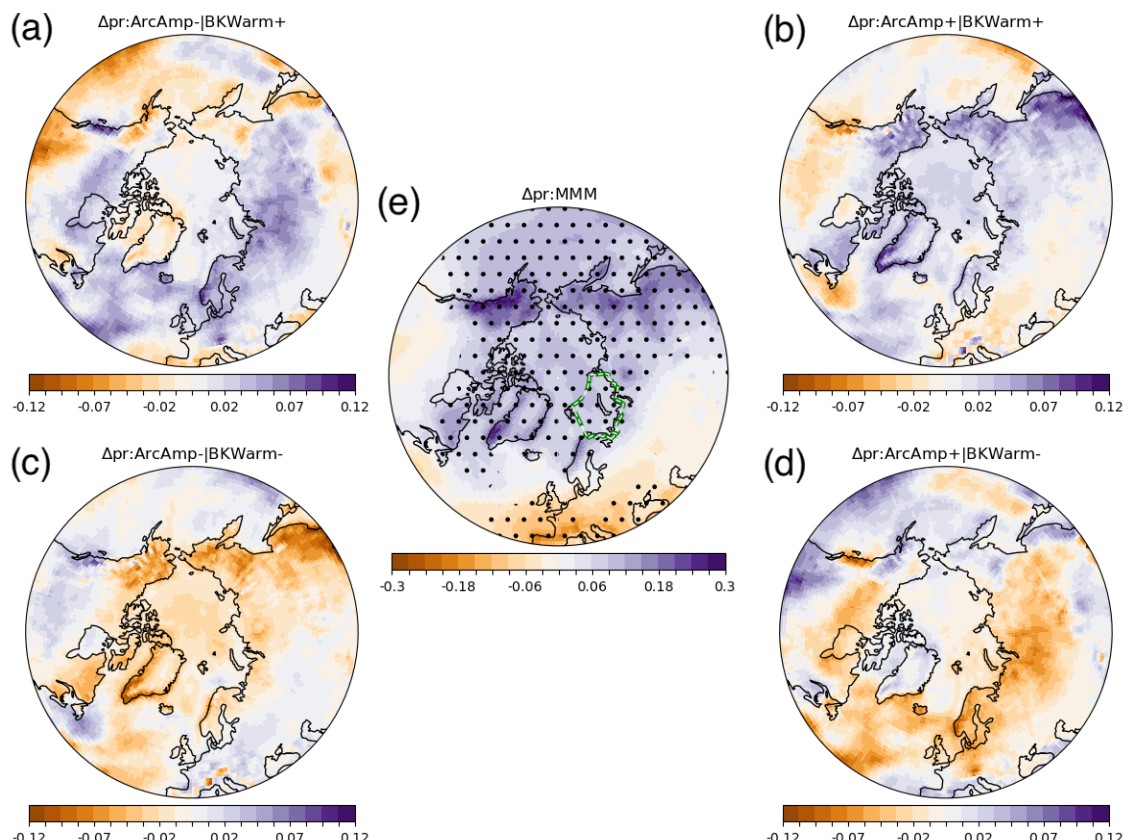

**Figure 5: Storylines of climate change for precipitation (a)-(d) and its MMM projection (e). Same convention as Fig. 3 applies.**

Figure 6 confirms the expected decline in sea-ice across the Arctic in the MMM, with sea-ice fraction displaying loss by at least 15% (cf. Fig. 6e). However, our storylines reveal a more complex picture than suggested by the MMM. On one hand, a Central Arctic amplification/dampening of these changes occur when BKWarm and ArcAmp changes are additive (Fig. 6b,c). On the other hand, large regional contrasts can appear when BKWarm and ArcAmp changes are of opposite sign (Fig. 6a,d): this is especially obvious when comparing the Atlantic and Pacific sector of the Arctic. Those changes appear to be associated largely with the Arctic atmospheric warming, with the Barents-Kara Sea warming playing a more local role with its effect being felt primarily in the Atlantic sector of the Arctic ocean.

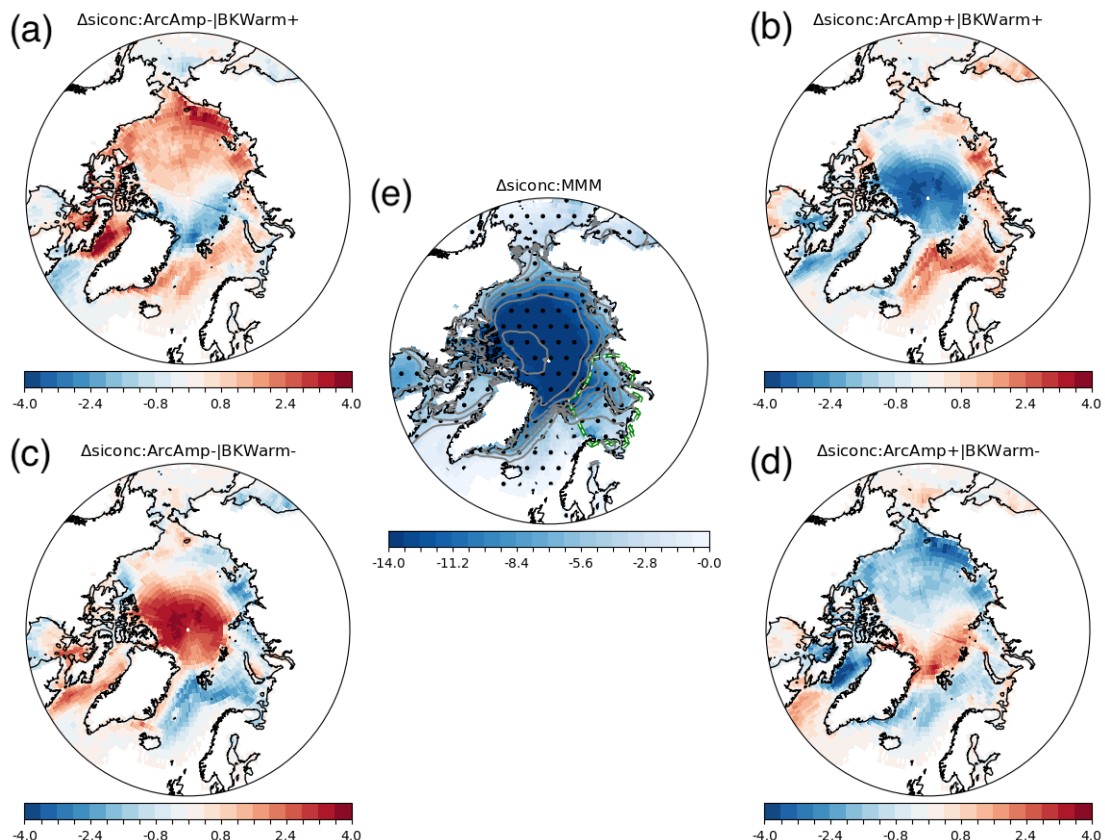

**Figure 6: Storyline of climate change for sea-ice fraction (a)-(d) and its MMM projection (e).**

## 4 Discussion and Conclusions

We produced four summertime climate change storylines for the Arctic region, for the four target variables that we consider to characterise seasonal change in the surface climate: 2 m temperature, precipitation rate, zonal wind at 850 hPa level, and sea-ice fraction over the Arctic region. We devised those storylines using an established methodology, previously applied to develop storylines across various midlatitude regions of both hemispheres (ZS17, ML20). We combined this framework with the realisation that Arctic climate change in summer is tightly associated with two climate indices, the Barents-Kara Sea warming (BKWarm) and Arctic atmospheric amplification (ArcAmp), which we used as predictors. Our choice of methodology and predictors was guided by two criteria: (i) our storylines should be representative of the diversity in model projections, and (ii) our predictors should be connected to physical processes. Criterion (i) ensures that the storylines capture a meaningful set of possible climate change realisations, while criterion (ii) allows for a scientific understanding of what drives this diversity in model projections. Criterion (i) is critical to the viewpoint of the end-users who need a plausible range of climate change scenarios, for instance to develop mitigation strategies, while criterion (ii) is of greater interest to scientists who desire insights regarding the drivers of climate change in the Arctic. When based on those two criteria, storylines can be

used to study possible impacts of climate changes, as well as categorise climate models by storylines; as such storylines are an
efficient way of identifying a few climate models most representative of the diversity of CMIP6 projections.

Our storylines are particularly successful at capturing the spread in model projections for the 2 m temperature: our primary
finding is the differential warming rates between terrestrial and marine areas, which we find to be a major source of divergence
in model projections. We also applied our storyline analysis to other variables, to a varying degree of success: the relevance
of storylines to each target variable must be assessed case-by-case, as different target variables may be controlled by distinct
processes. Likewise, our predictors are less successful at capturing changes in seasons other than the extended boreal summer.
The specificity of storylines to variables, seasons and regions is an important limitation of this methodology, as it relies on
careful tuning to comprehensively represent changes.

Using this methodology, we produced the four Arctic climate change: ArcAmp- / BKWarm+ (A), ArcAmp+ / BKWarm+ (B),
ArcAmp- / BKWarm- (C), ArcAmp+ / BKWarm- (D). Our storylines show noticeably different paths for Arctic climate
change, which deviate substantially from the multi-model ensemble mean. Compared to the MMM, cooler surface temperature
in storylines A and C suggests fewer fire risks and less extensive permafrost thawing, if undergoing the same amount of global
warming. Storylines B and D present the opposite outcome, with more intense land warming that may lead to greater fire risks
and more permafrost thawing. Concomitant changes in precipitation rates and surface wind are expected to modulate those
trends: for instance, a wetter summer could imply a reduced fire risk in storyline B compared to D, even if both storylines
show similar rates of warming over land. The combined impacts of physical changes at the surface on climate risks such as
fires and permafrost thaw can only be evaluated with a quantitative analysis that is beyond the scope of our study. Furthermore,
our analysis also shows that enhanced risks over land may or may not translate into enhanced impacts over marine areas. For
instance, storyline A--which showed a lessening of climate risks over land---is tied to an enhanced warming of the Arctic
Ocean and an amplified loss in sea-ice cover, suggesting a more navigable Arctic Ocean and greater disruptions in marine
primary production compared to the MMM. Beyond changes that may be consistent across the entire Arctic, storylines also
suggest futures in which regional contrasts are enhanced. For instance, storylines A and D show sea-ice cover shrinking may
have pronounced differences between the Pacific and Atlantic sectors of the Arctic Ocean; such changes would likely entail
regional differences in the volume of Arctic shipping or marine primary production. Overall, we demonstrate that storylines
can be used to better understand the range of possible climate outcomes for the Arctic that emerge from coupled climate
models, a critical step toward planning for climate adaptation strategies.

## Appendix A: Empirical storylines

We also tested an empirical method for producing storylines, in which predictor indices emerge from a principal component analysis (PCA). This is achieved by finding the first two components of a PCA applied to each target variable (von Storch and Zwiers, 2002), and using those as predictors. Specifically, we can express changes in a target variable $\Delta Z$ as:

$$\Delta Z(x,m) = \overline{\Delta Z}(x) + \sum_{i=1}^{N} EOF_i(x)\, PC_i(m) \qquad \text{(A1)}$$

Here, $EOF_i$ is the eigenmode and $PC_i$ the eigenvalues of the i-th mode, and the summation is done over $N$ principal components. As in the MLR storylines (Eq. 1), the PCA storylines describe the inter-model variability in model projections, that is with respect to the MMM changes. Comparing the two frameworks, we find that eigenmode $EOF_i(x)$ in Eq. (A1) is analogue to coefficient $\beta_i(x)$ in Eq. (1), and $PC_i(m)$ in Eq. (A1) to climate predictor $\Delta \widehat{P}_i(m)$ in Eq. (1). Following the same methodology to the physical storylines, we produce four "empirical" storylines:

$$\widehat{\Delta Z}_{+,+} = s(+EOF_1(x) + EOF_2(x)) \qquad \text{(A2a)}$$

$$\widehat{\Delta Z}_{+,-} = s(+EOF_1(x) - EOF_2(x)) \qquad \text{(A2b)}$$

$$\widehat{\Delta Z}_{-,+} = s(-EOF_1(x) + EOF_2(x)) \qquad \text{(A2c)}$$

$$\widehat{\Delta Z}_{-,-} = s(-EOF_1(x) - EOF_2(x)) \qquad \text{(A2d)}$$

As in Eq. (3), $s$ defines the standardised climate response in Eq. (A2), which is derived from a Chi-square distribution for 2 degrees of freedom and evaluated on the edge of the 80% confidence boundary region ($s = 1.26$). Compared to the 2-predictors MLR storylines (Eq. 3), the 2-components PCA storylines (Eq. A2) will better discriminate the spread in model projections, since the variance explained by the first two components of a PCA maximises the variance that can be explained in the intermodel spread from any two predictors. While PCA predictors present the advantage of being strictly orthogonal to each other by construction, they are not directly relatable to specific climate indices or physical processes, which is a substantial drawback for interpreting changes. For these reasons, empirical storylines may be useful for providing a representative range of climate outcome to end-users (perhaps even more so than the MLR storylines, if judging solely from the amount variance explained); however, they are likely to be less relevant for understanding the underlying processes driving the diversity in climate outcomes.

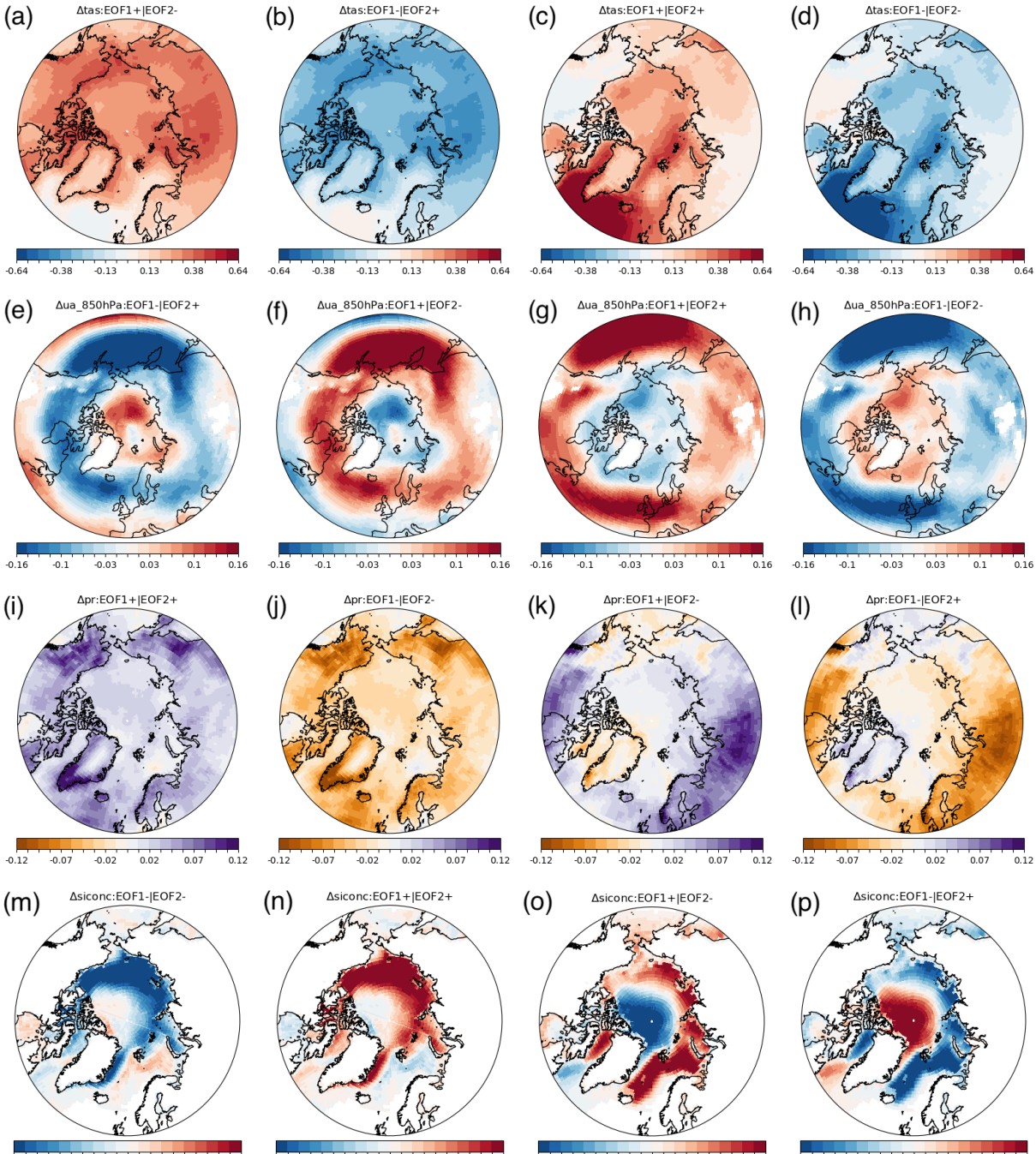

**Figure A1: EOF Storyline of climate change for: 2 m temperature, 850 hPa zonal wind, precipitation, and sea-ice fraction.**

Empirical storylines show qualitative similarities with the storylines presented in our study (see Fig. A1) to those found in our physical storylines for most target variables (Fig. 3-6), even if physical storylines consistently underperform empirical ones

with regards to the amount of explained variance in model projections. This is particularly true for the 2 m temperature, which
shows very similar patterns between empirical storylines and our storylines (compare Fig. A1 and 3).
**Appendix B: Optimizing Arctic storylines' predictors**
We selected the predictors for our Arctic storylines based on their ability to represent changes in key surface climate variables
in a linear regression framework. This entails that our two predictors should maximize the variance explained by the MLR
model while being as weakly correlated as possible (orthogonality of predictors is not strictly necessary but remains convenient
for interpreting changes). We already motivated in Section 2.4 that lower tropospheric temperature change (represented by
ArcAmp) and sea surface warming at high latitudes (represented by BKWarm) are the most relevant factors for defining our
two predictors; however, we did not explain the specific choices of pressure level or area for evaluating ArcAmp or BKWarm.
Table B1 shows the variance explained by the MLR model when using as predictors BKWarm (as defined in 2.4) and ArcAmp
(as defined in 2.4 but using the pressure level value shown in top row); Table B1 also shows the correlation coefficient between
BKWarm and ArcAmp at various levels.

|  | 1000 hPa | 925 hPa | 850 hPa | 700 hPa | 600 hPa | 500 hPa |
|---|---|---|---|---|---|---|
| Explained variance ($R^2$) | 0.40 | 0.43 | 0.41 | 0.35 | 0.32 | 0.30 |
| Predictors correlation ($r^2$) | 0.38 | 0.14 | 0.08 | 0.05 | 0.06 | 0.09 |

**Table B1: (top row) Explained variance for the 2-m air temperature over the Arctic by the multivariate linear regression model,**
**using BKWarm and ArcAmp as predictors, for various evaluation levels of ArcAmp. (bottom row) Correlation R2 of BKWarm with**
**ArcAmp, for various levels of evaluation of ArcAmp (columns) ranging from the lowest model level (1000 hPa; leftmost column) to**
**the mid-troposphere (500 hPa; rightmost column).**
Compared with other vertical levels, Table B1 shows that temperature at the 850 hPa level is only weakly correlated with the
Barents-Kara sea warming (0.08, see bottom row in Table B1) and also nearly maximizes the MLR explained variance (0.41,
see top row in Table B1). Specifically, MLR explained variance is found to decreases from a maximum value of 0.43 at 925
hPa to lower values higher in the troposphere, while the predictor correlation decreases swiftly above the lowest tropospheric
level (1000 hPa), which makes the 825 hPa level a reseasonable choice for defining ArcAmp. We also note that the 825 hPa
pressure level was selected to define Arctic Amplification in past studies (e.g., Manzini et al., 2014; ZS17).
Similarly to Table B1, we compare the variance explained by the MLR model and correlation coefficient when using as
predictors ArcAmp (as defined in 2.4) and sea surface warming averaged over various areas of the Northern Hemisphere
(including the Barents-Kara sea), as shown in Table B2. In addition to the Barents-Kara sea, we tested the Central Arctic and

North Atlantic ocean warming because of their covariability with Barents-Kara sea warming (Fig, 2a) and being areas where
intermodel variability in sea surface warming is the strongest at high latitudes (Fig. A1, a-d).

| | Barents-Kara Sea | Central Arctic Ocean | North Atlantic Ocean |
|---|---|---|---|
| Explained variance ($R^2$) | 0.41 | 0.40 | 0.45 |
| Predictor correlation ($r^2$) | 0.08 | 0.09 | 0.13 |

**Table B2: same as Table 1 but using various oceanic regions for our 'BKWarm' predictor: Barents-Kara sea (left column; [65°N, 80°N, 26°E, 95°E]; ocean only), Central Arctic ocean (middle column; [70°N, 90°N, 180°W, 180°E]; ocean only), North Atlantic ocean (right column; [45°N, 60°N, 70°W, 0°]; ocean only).**

Table B2 shows similar values for the MLR explained variance and predictor correlation when selecting either Central Arctic, North Atlantic or Barents-Kara sea warming. Based on this criterion alone, any of those three region could have been chosen as predictors for our Arctic storylines. Ultimately, we selected the Barents-Kara Sea as the reference area for defining our predictor because of its mediating role between the North Atlantic and the Arctic Ocean warming (e.g. Smedsrud et al., 2013), as explained in Section 2.4.

**Appendix C: storyline patterns - including the multi-model mean change**

Nearly all studies using the storyline approach show the total storyline patterns (e.g. ZS17), which corresponds to the response of the target variables to each predictor added upon the multi-model mean (MMM) change. Showing the full response is most relevant to the end-users to study climate risks but can make it more challenging to distinguish what differentiate storylines, because storylines' patterns are strongly influenced by the common MMM change. For convenience, we provide the total storyline patterns, defined by adding the MMM change (normalized by the global and annual-mean 2-m air temperature) to the storyline pattern defined in Eqns. 3a-d and shown in Figs. 3-6, for: 2-m air temperature (Fig. C1, a-d), 850 hPa zonal wind (Fig. C1, e-h), precipitation rate (Fig. C1, i-l), and sea-ice fraction (Fig. C1, m-p). We comment on what differs between storylines in Sections 3-4.

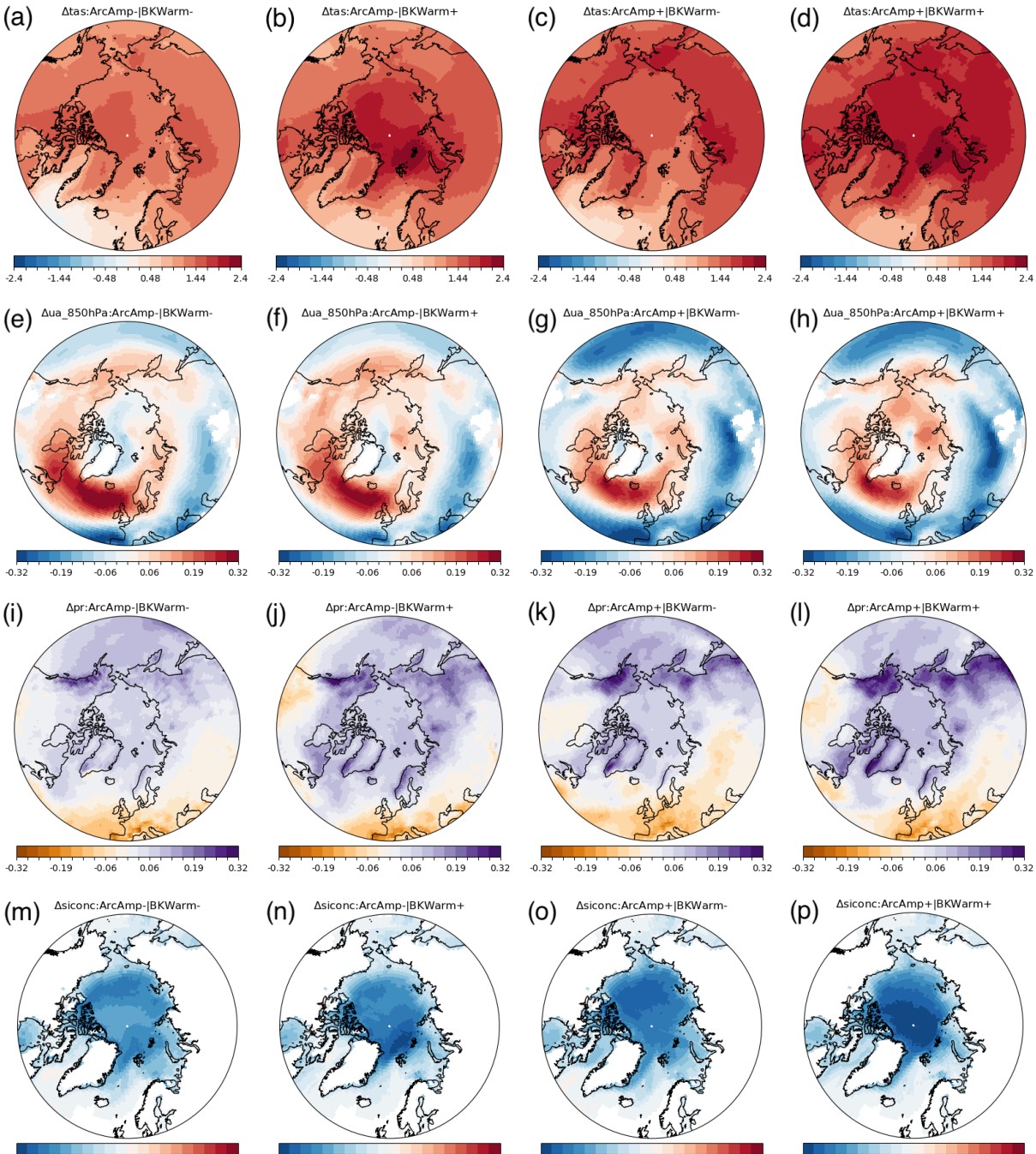

Fig. C1: "Overall storylines" of climate change for 2-m temperature, 850 hPa zonal wind, precipitation and sea-ice fraction. "Overall storylines" are defined by combining the multi-model ensemble mean change (Figs. 3-6, e) with our climate change storylines, as defined in Equation 3 and with patterns shown on panels a, b, c, d of Figures 3-6.

**Code availability**

The code to generate our Arctic storylines can be found on the first author's GitHub page (https://github.com/xlevine/Storylines_Analysis_ESD ).

**Data availability**

This study was based on World Climate Research Programme (WCRP)'s CMIP6 archived simulations, which can be found on The Earth System Grid Federation (ESGF). This data was stored locally on the National Infrastructure for Research Data (NIRD), a component of the Norwegian research infrastructure services (NRIS).

**Author contribution:**

XL performed the formal analysis and was responsible for the data presentation, supervised efforts leading to this work, and was responsible for the preparation of the manuscript. XL, RW, GM, AO, LG, DH were instrumental in setting the main goals and structure of this study and setting the storyline methodology. NJ, HL, LN provided important methodological inputs related to storylines' impact. RW, GM, AO, LG, DH, AK, RK, RW, NJ, HL, LN, PM helped with the preparation of this draft, providing critical comments. PM procured funding necessary to conduct this study and set the overarching goals of the PolarRES project that led to this study.

**Competing interests:**

The authors declare that they have no conflict of interest.

**Acknowledgements:**

We acknowledge the support of PolarRES (grant number 101003590), a project of the European Union's Horizon 2020 research and innovation programme. Storage and computing resources necessary to conduct this analysis was provided by Sigma2 — the National Infrastructure for High Performance Computing and Data Storage in Norway (project NS8002K and NN8002K). The CMIP6 simulations used for this analysis were obtained from the Earth System Grid Federation (ESGF), an infrastructure supported by the World Climate Research Programme (WCRP).

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
