# Peer review of "Storylines of Summer Arctic climate change constrained by Barents-Kara Sea and"

_EGUsphere, 2023_

## Author Comment (AC2)

**Figures and Tables for Reply on RC2 for EGUSPHERE-2023-2741**

[Figure]

**Figure 1**: (left) summer (May to October) and (right) winter (November to April) temperature change, averaged over the Arctic region (poleward of 55°N) and normalized by the global and annual-mean 2-m temperature change, for a large ensemble of CMIP6 models (unit: [K K$^{-1}$]). Both panels compare the climatology of the 2070-2099 period in the SSP5-8.5 scenario with the 1985-2014 period in the historical experiment. The solid black line shows the multi-model mean values.

[Figure]

**Figure 2**: Correlation coefficient of temperature averaged over the Arctic with Arctic 2-m temperature over (left) ocean regions only and (right) land regions only.

[Figure]

**Figure 3**: Response coefficient of 2-m temperature to (left) Arctic atmospheric warming and (right) BK seas warming, in summer. Note the lack of overlap in the climate response between land and ocean regions in their response to both predictors.

[Figure]

**Figure 4**: Response coefficient of 2-m temperature to (left) Barents-Kara seas warming (BKWarm) and (right) Arctic atmospheric warming (ArcAmp), in the summer season (May to October) (units: K K⁻¹).

|  | 1000 hPa | 925 hPa | 850 hPa | 700 hPa | 600 hPa | 500 hPa |
|---|---|---|---|---|---|---|
| Explained variance | 0.50 | 0.58 | 0.56 | 0.50 | 0.48 | 0.48 |
| Predictors Correlation | 0.55 | 0.29 | 0.16 | 0.14 | 0.16 | 0.22 |

**Table 1**: (top row) Explained variance for the 2-m air temperature over the Arctic by the multivariate linear regression model, using BKWarm and ArcAmp as predictors, for various evaluation levels of ArcAmp. (bottom row) Correlation $R^2$ of BKWarm with ArcAmp, for various levels of evaluation of ArcAmp (columns) ranging from the lowest model level (1000 hPa; leftmost column) to the mid-troposphere (500 hPa; rightmost column).

|  | Barents-Kara | Central Arctic | North Atlantic |
|---|---|---|---|
| Explained variance | 0.56 | 0.51 | 0.52 |
| Predictors Correlation | 0.16 | 0.13 | 0.12 |

**Table 2**: same as Table 1 but using various oceanic regions for our 'BKWarm' predictor: Barents-Kara sea (left column; [65°N, 80°N, 26°E, 95°E]; ocean only), Central Arctic ocean (middle column; [70°N, 90°N, 180°W, 180°E]; ocean only), North Atlantic ocean (right column; [45°N, 60°N, 70°W, 0°]; ocean only).

---

## Author Response (AR1)

**Summary of changes** to "Storylines of Summer Arctic climate change constrained by Barents-Kara Sea and Arctic tropospheric warming for climate risks assessment" by Levine et Co-authors.

Please find the revised version of my manuscript (revised file and marked-up manuscript version), which has been changed to reflect the helpful and constructive suggestions made by the two reviewers. Please note that, in the marked-up manuscript version, the strikeout text indicates removal of content, while the text in magenta font shows added content (normal black font is unchanged text since the original submission).

Because I have already responded to each of the reviewers' comments in the discussion section on the EGUSphere, I will not repeat the point-by-point reply to the reviewers (for any specific points raised by the reviewers, please refer to my replies in the EGUSphere discussion section). Instead, I am describing below the key changes made to the manuscript :

1. I've added three new models (CAMS-CSM1-0, FGOALS-g3, GISS-E2-2-G). We also removed two models (ACCESS-ESM1-5, EC-Earth3-Veg-LR); for ACCESS-ESM1-5, this was due to missing pressure levels in the raw output, and for EC-Earth3-Veg-LR because we found no significant differences with EC-Earth3 runs in the climate response and found it redundant.

2. We corrected a number of issues that were affecting a small subset of the CMIP6 simulations shown in our analysis (primarily stemming from issues with the raw CMIP6 data). These corrections have led to minor changes in figures 2-6 and table 1, which we detail below:
    a. In Table 1, MLR explained variance and predictors' correlation coefficient values have been modified. Changes in the values are generally minor, although those values are somewhat lower than previously stated. Compared to the first version of our draft, the most noticeable changes are for the MLR explained variance in the 2-m temperature and sea-ice fraction changes, which are noticeably lower than in the original draft. Yet, we note that none of those changes affect the main results; in particular, MLR explained variance for 2-m temperature still remains large (about ⅔ of the theoretical maximum), which validates the MLR model presented in our study for producing our Arctic storylines.
    b. Response patterns (Fig. 2a) and storylines patterns (Fig. 3-6) have incurred minor changes. The most substantial changes are the lower significance in the response of the 850 hPa zonal wind to BKWarm, and lower significance in sea-ice fraction changes to both BKWarm and ArcAmp. We note that those changes have not impacted our results and are still consistent with our findings as described in Section 3 and 4.

3. In response to Reviewer #1's comments, we added a third row to table 1, which repeats the analysis shown in the first row but for the regionally averaged values of our four

target variables. This row confirms Reviewer #1's idea, which is that changes are better captured on a regional than local scale.

4. We added 2 appendices: Appendix B provides an analysis supporting our choice of pressure level for ArcAmp and area for BKWarm; Appendix C shows the total storylines (i.e. the multi-model mean response added to storylines' pattern shown on Fig. 3-6).